# Delivery of a Hepatitis C Virus Vaccine Encoding NS3 Linked to the MHC Class II Chaperone Protein Invariant Chain Using Bacterial Ghosts

**DOI:** 10.3390/biomedicines12030525

**Published:** 2024-02-26

**Authors:** Yulang Chi, Shikun Zhang, Shouping Ji

**Affiliations:** 1College of Oceanology and Food Science, Quanzhou Normal University, Quanzhou 362000, China; ylchi@qztc.edu.cn; 2Department of Stem Cell and Regenerative Medicine, Institute of Health Service and Transfusion Medicine, Beijing 100850, China

**Keywords:** bacterial ghost, delivery DNA vaccine, transfection, immune response, hepatitis C virus NS3 protein, MHC class II chaperone protein invariant chain

## Abstract

Efficient delivery of a DNA plasmid into antigen-presenting cells (APCs) is a potential strategy to enhance the immune responses of DNA vaccines. The bacterial ghost (BG) is a potent DNA vaccine delivery system that targets APCs. In the present work, we describe a new strategy of using *E. coli* BGs as carriers for an Ii-linked Hepatitis C Virus (HCV) *NS3* DNA vaccine that improved both the transgene expression level and the antigen-presentation level in APCs. BGs were prepared from DH5α cells, characterized via electron microscopy and loaded with the DNA vaccine. The high transfection efficiency mediated using BGs was first evaluated in vitro, and then, the immune protective effect of the BG-Ii-*NS3* vaccine was determined in vivo. It was found that the antibody titer in the sera of BG-Ii-*NS3*-challenged mice was higher than that of Ii-*NS3*-treated mice, indicating that the BGs enhanced the humoral immune activity of Ii-*NS3*. The cellular immune protective effect of the BG-Ii-*NS3* vaccine was determined using long-term HCV *NS3* expression in a mouse model in which luciferase was used as a reporter for HCV *NS3* expression. Our results showed that the luciferase activity in BG-Ii-*NS3*-treated mice was significantly reduced compared with that in Ii-NS3-treated mice. The CTL assay results demonstrated that BG-Ii-*NS3* induced a greater NS3-specific T-cell response than did Ii-*NS3*. In summary, our study demonstrated that BGs enhanced both the humoral and cellular immune response to the Ii-*NS3* DNA vaccine and improved its immune protection against HCV infection.

## 1. Introduction

Persistent HCV infection is a global health problem with an estimated 71·1 million individuals chronically infected worldwide, accounting for 1% of the population. About 10–20% of them develop complications, such as cirrhosis, end stage liver disease, and hepatocellular carcinoma over a period of 20–30 years [1]. The nonstructural protein 3 (NS3) is one of the conserved HCV antigens and is associated with viral clearance. *NS3*-based DNA vaccines that may provide broad protection are potential candidates for an HCV vaccine [2,3,4]. However, the routine implementation of DNA vaccines is hindered by their poor immunogenicity and the requirement for high plasmid dosages. 

Antigen-presenting cells (APCs) play a crucial role in antigen-specific immune responses and attempt to enhance the potency of NS3-based DNA vaccines that focus on improving antigen gene expression, processing, and presentation to APCs have proved to be effective therapeutic strategies [5]. For example, employing the MHC class II-associated invariant chain (Ii) to improve antigen presentation is a potential approach to enhance the potency of DNA vaccines [6]. Additionally, linking the antigen to the class II-associated invariant chain peptide (CLIP) was reported to enhance antigen presentation via the MHC class II pathway [7,8,9]. Previously, we constructed an HCV DNA vaccine by replacing CLIP with the HCV NS3 CD4+ T helper 1 (Th1) epitope (residues 1248 to 1261) and found that linking the vaccine to the murine invariant chain (Ii) increased the antigen presentation to MHC class II molecules, resulting in enhanced, broadened, and prolonged CD8+ T cell responses [10,11]. However, the Ii-*NS3* plasmid must be delivered to and expressed in the APCs before the antigen can be presented.

Viral vectors have been reported to be the most potent carriers for gene delivery that target APCs [12,13]. Holst et al. previously developed an Ii-linked HCV *NS3* vaccine by using adenovirus as a delivery vector, which improved the CD8+ T cell-mediated cytotoxicity [14]. However, the immune responses against vector-derived antigens could reduce the efficacy and stability of viral vaccines and cause significant side effects. Furthermore, the high cost, low DNA-carrying capacity, and the risk of causing gene mutations limits the applications of viral vectors [15,16]. Hence, the aim of this study was to find a safe carrier for *NS3*-based DNA vaccine delivery.

Non-viral vectors are promising gene delivery systems with low cytotoxicity, immunogenicity and mutagenesis. However, they do not carry ideal characteristics and have faced critical challenges, including gene transfer efficiency, specificity, gene expression duration, and safety [17]. The bacterial ghost (BG) platform system is a novel DNA vaccine delivery system. BGs maintain the cellular morphology and antigenic structures of native bacteria. Thus, loading BGs with recombinant DNA takes advantage of the excellent bioavailability of DNA-based vaccines and the high expression rates of the DNA-encoded antigens in APCs, such as macrophages and dendritic cells. Additionally, like living bacteria, BGs act as a natural adjuvant for the recruitment of innate immunity master regulators and the activation of APCs [18,19,20]. Moreover, BGs are non-living empty bacterial envelopes, which make them safer than living bacteria.

Based the above information, we propose a new strategy of using BGs as delivery carriers for the Ii-linked *NS3* DNA vaccine. This strategy will take advantage of the excellent properties of BGs, which include targeting APCs and being natural adjuvants to improve the expression level of the DNA vaccine in APCs such as macrophages and dendritic cells. Moreover, the exogenous expressed Ii-linked NS3 peptides can be directly delivered to the MHC class II molecules, and therefore, the antigen presentation of NS3 peptides will be enhanced. In this study, *E. coli* BGs were prepared and used for loading the DNA vaccine. The efficiency of the gene delivery of the BG-mediated Ii-linked *NS3* plasmid was studied in APCs and the protective immunity of the DNA vaccine was evaluated in a murine challenge model.

## 2. Materials and Methods

### 2.1. Materials and Ethics Declaration

Dulbecco’s modified Eagle’s medium (DMEM Cat. No 11965092), RPMI-1640 medium (Cat. No 72400146), fetal bovine serum (FBS, Cat. No 10099158), Opti-MEM medium (Cat. No 31985070) and MitoTracker™ Green (Cat. No M46750) were purchased from Invitrogen Life Technologies (Carlsbad, CA, USA). 3-[4,5-Dimethylthiazol-2-yl]-2,5-diphenyltetrazolium bromide (MTT, Cat. No 475898) was obtained from Sigma-Aldrich (St Louis, MO, USA). The luciferase assay system (Cat. No E1501) and the pGL3 plasmid (Photinuspyralis luciferase under the control of the SV40 promoter, Cat. No E1751) were obtained from Promega (Madison, WI, USA). Branched Polyethylenimine (BPEI) (MW 25 kDa, Cat. No 23966-1) was purchased from Polysciences Inc. (Warrington, PA, USA) and dissolved in phosphate-buffered saline (PBS; pH 7.4) at 5 mg/mL as stock solution. The study is reported in accordance with ARRIVE guidelines (https://arriveguidelines.org, accessed on 20 February 2024). All experimental protocols were approved by the Quanzhou Normal University Ethics Committee.

### 2.2. Preparation of E. coli Bacterial Ghosts

DH5α *E. coli* BGs were prepared as previously described [21]. Briefly, DH5α cells transformed with pHH43, a heat-inducible plasmid containing the E lysis gene, were cultured in LB medium at 28 °C until the OD_600_ of the medium reached 0.5 and the temperature was increased to 42 °C. After the cells were completely lysed, the BGs were harvested, lyophilized and stored until further use. 

### 2.3. Plasmid Construction

DNA fragments to encode the HCV-NS3 Th1 epitope from residue 1248 to residue 1261 (5′-AATTC ATG GGC TAC AAG GTG TTG GTG CTC AAC CCC TCT GTT GCT GCA ACG T-3′) were synthesized using Invitrogen. The sense and antisense strands were annealed at 95 °C for 5 min followed by slow cooling to room temperature. The annealed fragment was digested with *Eco*RI and *Xba*I and subcloned into the pVAX1 vector (Invitrogen Life Technologies). The recombinant plasmid was named pVAX1-*NS3*.

The murine Ii p41 cDNA [22] (a kind gift from Dr. Ronald N. Germain, National Institutes of Health, Bethesda, MD, USA) was subcloned into the *Hin*dⅢ/*Xho*I sites of pVAX1 to create pVAX1-Ii (Figure 1A). The partial mIi DNA fragment (between the *Hin*dIII and *Acl*Ⅰsites), in which the CLIP fragment was replaced by *NS3* Th1, was synthesized using Invitrogen. The sequence of the fragment containing the *Hin*dIII and *Acl*Ⅰsites (underlined) and the *NS3* Th1 epitope (lower case) is 5′-AAGCTT ATG ACG GAT CCG CAT GCG AGC TCG GTA CCC CAG CTT CCG AAA TCT GCC AAA CCG GTG AGC CAG ggc tac aag gtg ttg gtg ctc aac ccc tct ggt gct gca acg CGG CCG ATG TCC ATG GAT AAC ATG CTC CTT GGG CCT GTG AAG AACGTT-3′. The synthesized fragment and pVAX1-Ii were digested with *Hin*dIII and *Acl*Ⅰ and ligated into the pVAX1-Ii plasmid between the *Hin*dIII and *Acl*Ⅰ sites to construct the Ii-linked *NS3* plasmid (namely pVAX1-Ii-*NS3* Th1) in which the CLIP DNA fragment was substituted by the *NS3* Th1 DNA fragment (Figure 1B). 

### 2.4. Loading the BGs with the pVAX1-Ii-NS3 Th1 Plasmid

Membrane vesicles of DH5α were prepared as described [23]. Briefly, the DH5α cells were collected and treated with lysozyme at 30 °C for 30 min followed by ultrasonication for 30 min. The membrane vesicles were washed via centrifugation (285,000× *g* for 60 min) and resuspended in Tris-acetate buffer, then stored at −70 °C for further use.

To load the plasmid, the BGs and the PI-labeled pVAX1-Ii-*NS3* plasmid were dissolved in HBS (NaCl 100 mmol/L, sodium acetate 10 mmol/L, HEPES 10 mmol/L, pH 7.0), such that the final concentration of the BGs was 2 mg/mL and the concentrations of the plasmid ranged from 1.0 μg/mL to 10.0μg/mL. The mixture was incubated at 28 °C for 90 min, then membrane vesicles (2 mg/mL) and CaCl_2_ (25 mmol/L) were added and incubated overnight at 37 °C. The BGs were then labeled with 0.25 μg/mL of MitoTracker Green FM (Molecular Probes, Leiden, The Netherlands) at 37 °C for 15 min [24]. The loaded bacterial ghosts were analyzed using confocal microscopy [25] (Radiance 2100TM; Bio-Rad, Hercules, CA, USA) using an inverted fluorescent microscope (TE300; Nikon, Melville, NY, USA). The excitation and emission wavelengths were 488 nm and 543 nm, respectively. The samples were also examined using transmission electron microscopy (Philips, Eindhoven, The Netherlands) and scanning electron microscopy (Hitachi, Tokyo, Japan), as well as through flow cytometry (Becton Dickinson, Palo Alto, CA, USA).

### 2.5. Transfection Experiments

DC2.4 human dendritic cells and Raw264.7 human macrophage cells obtained from ATCC (Manassas, VA, USA) were grown in RPMI-1640 medium supplemented with 10% fetal bovine serum (FBS) and DMEM medium supplemented with 10% heat-inactivated FBS, respectively. The cultures were maintained at 37 °C in a humidified 5% CO_2_ environment. The cells were trypsinized and 500 μL of cells (1.6 × 10^5^/mL) were seeded in for each well of a 24-well plate 16 h before transfection. The medium was replaced by Opti-MEM medium and MitoTracker-labeled BGs loaded with pGL3 plasmid at different concentrations were added to cells. The cells were incubated for 2 h and the medium was replaced with complete medium for 48 h of incubation. The fluorescent signal of the MitoTracker Green labeled BGs within the cells was examined using confocal microscopy. RAW 264.7 cells were also transfected with naked pGL3 plasmid (negative control) or pGL3 plasmid loaded by BGs or BPEI (positive control), respectively.

To quantify the transfection efficiency, the MitoTracker signal in the transfected cells was determined through flow cytometry. The transfection efficiency of luciferase was determined with a luciferase assay system [26] and the relative light units (RLU) produced were measured with a 96 microplate luminometer (Glomax; Promega). The protein concentrations of the cell extracts were measured using the BCA assay. The final values were reported in terms of RLU/mg protein.

### 2.6. Mouse Immunization

Inbred female BALB/c (H-2^d^) mice that were 6–8 weeks old were obtained from the Experimental Animal Center of the Academy of Military Medical Sciences and were housed under SPF conditions at the National Beijing Center for Drug Safety Evaluation and Research (NBCDSER). Animals were treated according to the guidelines of NBCDSER and Beijing Institute of Transfusion Medicine. Forty-five female BALB/c mice were randomly divided into nine groups of five each to receive an injection of a naked vaccine (p*NS3*/4A, pTh1 or pIi-Th1), or a vaccine loaded in bacterial ghosts (p*NS3*/4A in bacterial ghosts, pTh1 in bacterial ghosts or pIi-Th1 in bacterial ghosts), or to be the experimental controls (PBS, pVAX-1 or bacterial ghosts). A 100-μL aliquot of plasmid at a concentration of 1 μg/μL was injected into the quadriceps of each mouse. All of the groups received a booster injection at one-week intervals. After the third injection, blood was collected from all of the mice and the serum was obtained.

### 2.7. Anti-NS3 ELISA 

Ninety-six-well ELISA plates (Nunc, Naperville, IL, USA) were coated with 100μL/well of 50 mM sodium carbonate buffer (pH 9.6) containing 1 μg/mL of NS3 synthesized by SBS Genetech Co (Beijing, China) and incubated overnight at 4 °C. The plates were then blocked using incubation with PBS buffer containing 1% defatted milk powder at 37 °C for 1 h. After washing five times with PBS/Tween, 100 μL of 200-fold diluted immunized mice serum was added to the wells for a 1 h incubation at 37 °C. The bound murine serum antibodies were detected via reaction of an alkaline phosphatase-conjugated goat anti-mouse IgG antibody (Zhongshan Biotechnology Co., Ltd., Beijing, China) with the substrate p-nitrophenyl phosphate. The reaction was stopped by adding 1 N NaOH, and the OD_450_ was determined.

### 2.8. A Reporter Mouse Model That Allows Noninvasive Detection of HCV NS3/4A Expression in the Liver

A long-term expression mouse model that specifically expresses both the HCV NS3/4A protein and the firefly luciferase in the liver was constructed using hydrodynamic injection in combination with phiC31 integrase as described previously [27]. Briefly, the phiC31 integrase plasmid and the reporter vector pAA-*att*B-*NS3*/4A-Fluc (containing an *att*B site and the genes for HCV *NS3*/4A and firefly luciferase under the control of the human alpha-antitrypsin enhancer/promoter (AATP) were co-delivered to murine livers using high-pressure tail vein injection. Because of the phiC31-mediated intramolecular recombination between the wild-type *att*B and *att*P sites in the mice, the *NS3*/4A and luciferase expression cassettes were permanently integrated in our mouse strain. The HCV *NS3*/4 expression would be reflected by the luciferase activity, which can be accurately monitored in vivo using bioluminescence imaging. 

### 2.9. Luciferase Activity Monitoring in Living Mice Using an IVIS Camera [28]

Ten minutes prior to imaging, 150 mg/kg of D-luciferin was intraperitoneally injected into the immunized mice. The mice were anesthetized with 1–3% isoflurane before imaging and continuously exposed to 1–2% isoflurane to maintain sedation. The luciferase activity was detected in vivo using the IVIS imaging system (Xenogen, Alameda, CA, USA), and the data were digitized and electronically displayed as a pseudocolour overlay onto a grey scale image of the animal. Regions of interest from displayed images were quantified as photons/s/cm^2^/sr using the Living Image 4.0 software.

### 2.10. Assessment of Specific Cell Lysis [29,30]

The standard lactate dehydrogenase release assay was performed according to the manufacturer’s instructions to detect the NS3-specific CTLs. SP2/0 cells that stably expressed HCV NS3 protein were used as target cells to test the peptide-specific CTLs. Splenocytes were prepared from the immunized mice and pulsed with the synthesized NS3 1248–1261 peptide for 30 min at 37 °C, then 1.5 × 10^4^ target cells were added at a ratio of 1:10 (target cells/splenocytes) in a final volume of 100 μL. The plates were incubated for 3 h and 15 min at 37 °C and then the cells were spun down. Ten microliters of 10× lysis buffer was added into the wells and incubation was continued at 37 °C for 45 min. Fifty microliters of supernatant was transferred to a fresh ELISA plate, 50 μL of substrate was added to each well and mixed. The mixture was incubated for 30 min in the dark at room temperature and 50 μL of stop solution was added to each well. The absorbance at 490 nm was read and the specific lysis was calculated using the following formula:% Cytotoxicity = 100 × (Experimental value − Spontaneous Effector value − Spontaneous Target value)/(Maximal Target value − Spontaneous Target value).

The endpoint titer of anti-NS3 antibody in immunized mice was calculated as the dilution at which OD450 of the serum sample fell below the cut-off.

### 2.11. Statistical Analysis 

The statistical significance of the differences among groups was determined using unpaired Student’s *t* tests.

## 3. Results 

### 3.1. Preparation and Characterization of the DH5α BGs

After induction at 42 °C, the OD_600_ of the pHH43-transformed DH5α bacteria increased to approximately 0.9 within 30 min, then dropped rapidly to 0.3 at approximately 1.5 h after induction, whereas the cells sustained growth at 28 °C (Figure 2A). These results indicated that the E lysis gene was induced by the elevated temperature. Bacterial ghosts (BGs) were prepared via centrifugation. The morphology of the BGs was examined using SEM and TEM. The BGs exhibited a relatively intact surface structure except for the lysogenic tunnels (Figure 2B). TEM revealed that the DH5α cells had an intact bacterial morphology with cytoplasm present (dark stain) (Figure 2C), whereas the DH5α BGs appeared to be cellular envelopes due to the loss of most of the cytoplasmic materials (Figure 2D). The membranes of the BGs appeared to be mostly intact but were sometimes detached from the bacterial cytoskeleton.

### 3.2. Determination of the Plasmid DNA Loading Efficiency of the BGs

The plasmid DNA loading efficiency of the BGs was then determined. MitoTracker-labeled *E. coli* BGs loaded with PI-labeled plasmid DNA were observed using confocal microscopy as well as transmission electron microscopy (TEM). As shown in Figure 3, PI-labeled plasmid DNA (red fluorescence) appeared in most of the MitoTracker-labeled *E. coli* BGs, indicating that the *E. coli* BGs were loaded with the pVAX1-Ii-*NS3* Th1 plasmid. The TEM images demonstrated that the BGs were transparent, whereas electron-dense substances were present in the plasmid-loaded BGs, which also indicated that the plasmid DNA was successfully loaded into the BGs.

The BG samples were also analyzed using flow cytometry (Figure 4). B1 had only red fluorescence, B2 had both red and green fluorescence, B3 had neither green nor red fluorescence, and B4 only ha green fluorescence. The results revealed that more than 98% of the BGs displayed both red and green fluorescence at the optimal condition, which indicated that nearly 100% of the MitoTracker-labeled BGs were loaded with PI-labeled pGL3 or the pVAX1-Ii-*NS3* Th1 plasmid.

### 3.3. Evaluating the Efficacy of BG-Mediated Transfection In Vitro

The efficacy of BG-mediated transfection was further investigated in vitro. The uptake of BGs loaded with the pGL3 plasmid by RAW 264.7 cells, a macrophage cell line, was investigated using fluorescence confocal microscopy. As shown in Figure 5A,B, almost all the macrophage cells exhibited green fluorescence from the MitoTracker-labeled BGs in their cytoplasm. Moreover, the fluorescence signal in these macrophages was determined using flow cytometry and compared to that of control cells. A fluorescence shift of the whole cell population relative to the control cells was observed (Figure 5C–E). The percentage of fluorescently labeled cells among the BG-treated cells was approximately 95%. These results suggested that the macrophages could uptake BGs with high efficiency.

The BG-mediated transfection efficiency was also evaluated using luciferase as a reporter gene. The activity of luciferase is represented in Figure 5D. RAW 264.7 cells transfected with BGs loaded with the pGL3 plasmid displayed high luciferase activity similar to that transfected with PEI, while cells transfected with naked pGL3 plasmid DNA exhibited background luciferase activity (Figure 5F). Taken together, these results indicated that BGs could target macrophage cells and deliver plasmid DNA via cellular uptake. 

### 3.4. Humoral Immune Responses following Immunization with Plasmid Vaccines Loaded in Bacterial Ghosts or Free Plasmids

Next, the immune protective effect of the BG-Ii-*NS3* vaccine was determined in vivo. Mice were inoculated with BGs loaded with the Ii-*NS3* plasmid (BG-Ii-*NS3*) or with naked Ii-*NS3* plasmids (Ii-*NS3*). As shown in Figure 6, the endpoint titer of antibodies to HCV-NS3 were calculated in the serum of mice inoculated with the naked Ii-*NS3* plasmid and Ii-*NS3* plasmid loaded in BGs. However, the anti-NS3 antibody endpoint titer of the BG-Ii-*NS3* group was 16 times than that of the Ii-*NS3* group, indicating that the BGs efficiently enhanced the immune response to the Ii-NS3 protein. In contrast, almost no antibodies were detected in the mice in the pVAX-1 and BG groups, indicating that the anti-NS3 antibody was specifically induced by the Ii-*NS3* plasmid. 

### 3.5. Evaluation of the Immunogenicity of the DNA Vaccine in the Mouse Model

To evaluate the capacity of the BGs to improve the immunogenicity of Ii-*NS3*, plasmid pAA-*att*B-*NS3*/4A-Fluc that contains minimal length ϕC31 *att*B site and the genes for HCV *NS3*/4A and firefly luciferase under the control of the human alpha-antitrypsin enhancer/promoter (AATP) was generated (Figure 7A), which enabled bioluminescent imaging of HCV *NS3*/4A expression in living mice. The plasmid pAA-*att*B-*NS3*/4A-Fluc together with Pphic31φ [31] were hydrodynamically injected into mice that had been immunized with BG-Ii-*NS3*, Ii-*NS3* or the empty vector. The luciferase activity was evaluated 1 d and 10 d post-injection using in vivo imaging. No significant difference was observed among these three groups of mice1 d post-injection (Figure 7B). The activity of luciferase 10 d post-injection was relatively low in the BG-Ii-*NS3* and Ii-*NS3* groups, approximately 2–5 times lower than that in the control group. Notably, BGs loaded with Ii-*NS3* were able to reduce markedly the luciferase activity in the BG-Ii-*NS3* group (by up to 50%) compared to the Ii-*NS3* group (Figure 7C), indicating that immunization with BGs enhanced the specific T-cell responses in living mice.

Further, we compared the capacity of BG-Ii-*NS3* and Ii-*NS3* to induce NS3-specific T-cell responses in vivo. A cytotoxicity assay was performed 2 weeks after the last immunization. Spleen lymphocytes from immunized mice were used as effectors and SP2/0 cells expressing NS3 were used as targets. The mice immunized with BG-Ii-*NS3* had higher level of cytotoxicity than did those immunized with the naked Ii-*NS3* (25% versus 17.4%) (Figure 7C), indicating that immunization with BG-Ii-*NS3* induced a higher CTL response specific for Ii-*NS3*.

## 4. Discussion

A DNA carrier that targets APCs is a powerful strategy for improving the immune response to a DNA vaccine. Viral vectors were the most efficient gene carriers until recently. The development of live bacteria as gene carriers is an exciting research area for DNA vaccine delivery [32,33,34,35]. The use of bacterial carriers as a delivery system allows the specific targeting of APCs and evades the host’s defense against exogenous DNA by having it enclosed by the bacteria, potentially allowing long-term antigen expression [36]. Bacterial carriers also act as a natural adjuvant that activates the APCs [18,37]. Bacteria can be attenuated by generating gene deletions that render the risk of virulence reversion almost negligible. However, the application of attenuated bacteria in nonnative environments is potentially pathogenic or may cause severe bacteremia [18,37]. These results demonstrate that flagellin can directly stimulate human but not murine DC maturation, providing an additional mechanism by which motile bacteria can initiate an acquired immune response.

It has been demonstrated that BGs preserve the antigenic nature of membrane components of native bacteria, including LPS, peptidoglycan or flagella [19,20]. Therefore, these envelope structures are efficiently recognized and taken up by immune and non-immune cells [38,39,40]. BGs stimulate cells through TLR2 and TLR4 pathways [41,42], which forms the basis of their adjuvant activity. DCs are unique APCs with abilities to prime naïve T cells, and thus play an essential role in the initiation of primary immune responses. BGs are efficiently taken up by DCs and result in the induction of proinflammatory cytokines, which subsequently upregulate the costimulatory molecules on DCs for efficient presentation of foreign antigens to naive T cells [43]. Research has shown that BGs deliver efficient and early maturation signals to the DCs, especially IL-12, which are the main cytokines driving NK and Th1 cell stimulation [38,43,44]. The MHC-II levels are upregulated after 12 h exposure to BGs [38,43], indicating that they have the potential to induce early protective immune responses, which are very much required during emergency vaccination. BGs also enhance MHC-I expression on DCs and the presence of LPS effectively improves the cross presentation and maturation of DCs [45,46]. These findings suggest that BGs have the ability to stimulate both humoral and cell mediated immune responses. Besides DCs, BGs also effectively stimulate monocytes and macrophages and polarize the response toward Th1 [47]. All these factors contribute to the overall potency of BG adjuvated vaccines. Felnerova et al. [48] also showed that BGs induce a proliferative response in T cells and this proliferation capacity was higher in the cultures including APCs than in cultures stimulated with BGs only. Thus, BGs activate T cells either directly through TLR or indirectly through the presentation of cognate antigen by APCs. 

Bacterial ghosts have most of the characteristics of ideal DNA vaccine delivery carriers [21,22,23,49], including a high capacity for loading plasmid DNA, specific targeting of the DNA-encoded antigens to the primary APCs and functionality as adjuvant particles to activate the APCs. Moreover, they are easy to produce, can be stored long-term at ambient room temperature as a lyophilized product, have low production costs and are safer than live bacteria. Additionally, BGs could be administered in various ways, such as via intramuscular injection, subcutaneous injection, mucosal delivery [21,50] and orally [51,52]. Some people are concerned that the injection of BGs may cause serious toxic side effects induced by endotoxins. However, Mader et al. [53] have shown that experimental animals have a tolerance level of lipopolysaccharides on the surface of a BG at least two orders of magnitude higher than free lipopolysaccharides, and the dosage of a BG used in the study does not cause endotoxin-related toxic side effects, thus not affecting its safety as a DNA delivery carrier [53]. The potency, safety, stability and relatively low cost of bacterial ghosts offer a significant technical advantage, especially when used for combination vaccines. Hence, the aim of this study was to deliver the Ii-linked *NS3* DNA vaccine using *E. coli* BGs as carriers to improve the antigen expression in the APCs and thus the immunogenicity of the *NS3* DNA vaccine.

The DH5α cells were cultured at 28 °C and disrupted by the induced expression of a plasmid-encoded lysis E protein at 42 °C. BGs were prepared through centrifugation and examined by electron microscopy. The BGs exhibited a relatively intact surface membrane structure except for the lysogenic tunnels. The BGs lacked cytoplasm, leaving room to load the plasmid DNA. To load the plasmid DNA into the BGs, plasmids were incubated with BGs and then the BGS were sealed with *E. coli* membrane vesicles. Both fluorescence confocal microscopy and electron microscopy observations indicated that the plasmid was indeed loaded into the BGs and that the plasmid DNA was loaded into the BGs with high efficiency under optimal conditions. Sealing the BGs with membrane vesicles is a very important step because otherwise the DNA subsequently leaked from the BGs. 

BG-mediated transfection was studied. The results showed that BGs were effectively taken up by the RAW264.7 macrophage cells so that nearly 100% of cells exhibited a BG signal. This result suggested that the BGs could efficiently target human macrophage cells. The luciferase activity in the RAW264.7 cells transfected with pGL3-loaded BGs was also determined, which indicated that the BGs delivered the plasmid to the macrophage cells with high efficiency. Together with the DNA loading results, these findings indicated that BGs may be a good gene delivery carrier in which to load plasmid DNA and target the plasmid DNA to the antigen-presenting cells.

The Ii-linked *NS3* DNA fragment was subcloned into the pVAX-1 plasmid and the recombinant plasmid was called pVAX1-Ii-*NS3*-Th1 (abbreviated as Ii-*NS3*). Naked Ii-*NS3* plasmids or Ii-*NS3* plasmids loaded in BGs (abbreviated as BG-Ii-*NS3*) were intramuscularly injected into mice three times at biweekly intervals. Serum samples were collected and the anti-*NS3* antibody levels were determined via an ELISA using the synthesized *NS3* peptide. The antibody titer in the sera of the BG-Ii-*NS3*-immunized mice was higher than that of the Ii-*NS3*-immunized mice, which indicated the BGs enhanced the humoral immune response to Ii-*NS3*, most likely by delivering the plasmid to the antigen-presenting cells and increasing the antigen expression level. 

The immune protective effect of Ii-*NS3* or BG-Ii-*NS3* was also evaluated in the long-term HCV-expression mouse model in which the HCV NS3 expression reflected by the luciferase activity can be accurately monitored in vivo using bioluminescence imaging. The luciferase activity in the mice injected with BG-Ii-*NS3* was approximately five times lower than that in the untreated mice, whereas the luciferase activity in the Ii-*NS3*-immunized mice was half that of the untreated mice. These results suggested that BGs efficiently improved the immune response to the Ii-*NS3* DNA vaccine directed against HCV. The CTL assay results demonstrated that the BG-Ii-*NS3* vaccine induced a higher NS3-specific T-cell response than Ii-*NS3* did. The results indicated that BGs significantly enhanced the NS3-specific T-cell response to the Ii-*NS3* plasmid product and improved the immune protection against HCV infection.

In summary, BGs were loaded with plasmid DNA with high efficiency and efficiently delivered the vaccine plasmid to the APCs, resulting in a high level of gene expression. Furthermore, BGs enhanced both the humoral and the cellular immune response to the Ii-*NS3* DNA vaccine and improved the immune protection against an HCV infection. Although the safety of *Escherichia coli* BG for human use has not been proven, however, other Gram-negative bacteria, such as Lactobacillus casei [54], attenuated Salmonella typhimurium [55], and attenuated Shigella flexneri 2a [56], have been proven to be safe as vaccines. Making these bacteria into ghosts for delivering DNA vaccines will further enhance their safety and is expected to be widely used as an example. Recently, the bacterial ghosts of the aforementioned strains have been prepared and proven to be able to effectively mediate immune responses and can be used as a delivery system for DNA vaccination [57,58,59]. We believe that the BGs would be of significant benefit in vaccine development, and our strategy can improve the immune response efficiency of other DNA vaccines.

## Figures and Tables

**Figure 1 biomedicines-12-00525-f001:**
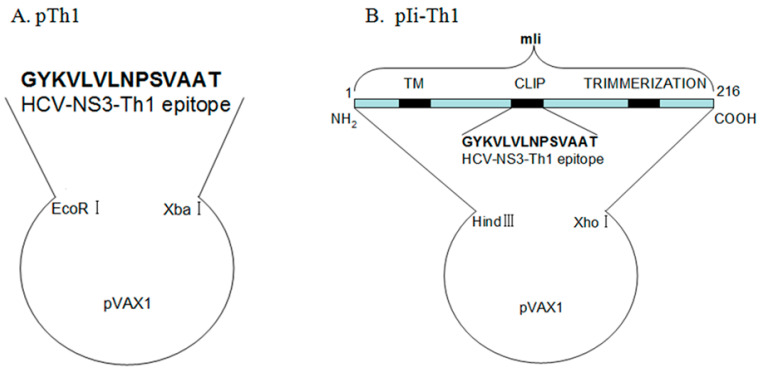
Schematic of the construction of the Ii-linked *NS3* plasmid. (**A**) The DNA fragment of the HCV-*NS3* Th1 epitope (amino acids 1248–1261) was subcloned into the *EcoRI/XbaI* sites of the pVAX1 plasmid. (**B**) The mIi-linked *NS3* was constructed by ligating the mouse invariant chain (mIi, 41 KDa) coding region with the CLIP coding region of mIi substituted by the HCV-*NS3* Th1 epitope into the *HindⅢ* and *XhoI* sites of the pVAX1 plasmid shown in (**A**).

**Figure 2 biomedicines-12-00525-f002:**
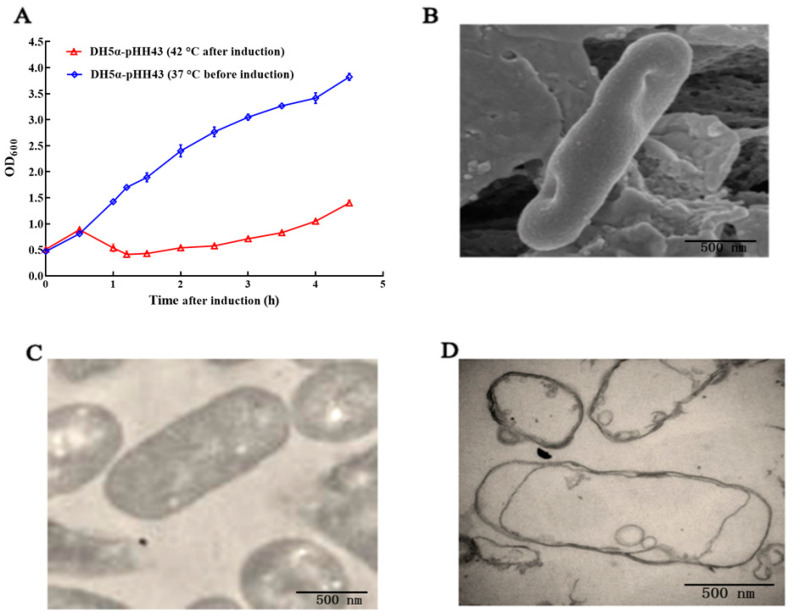
Preparation and morphological characterization of the DH5α BGs. (**A**) Induction of E protein-mediated cell lysis at 42 °C. (**B**) Scanning electron microscopy (SEM) of the DH5α BGs, (**C**) Transmission electron microscopy (TEM) of the DH5α cells, (**D**) TEM of the DH5α BGs. The magnification of the SEM and TEM images was 24,000×.

**Figure 3 biomedicines-12-00525-f003:**
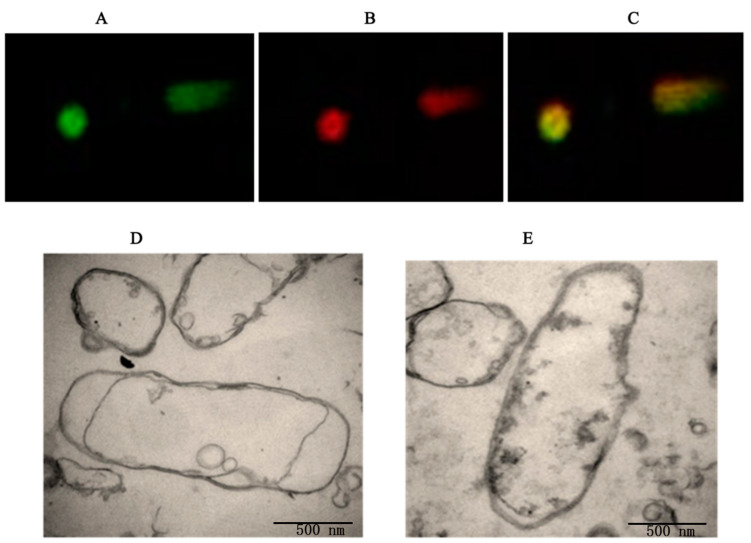
Fluorescence confocal microscopy and transmission electron microscopy of *E. coli* ghosts loaded with the pVAX1-Ii-*NS3* Th1 plasmid. (**A**–**C**), Fluorescence confocal microscopy images: (**A**) MitoTracker-labeled (green) *E. coli* ghosts, (**B**) PI-labeled plasmid, (**C**) Overlay of the differential interference contrast image and fluorescence images (**A**,**B**). (**D**,**E**) Transmission electron microscopy images of BGs and BGs loaded with plasmids, respectively.

**Figure 4 biomedicines-12-00525-f004:**
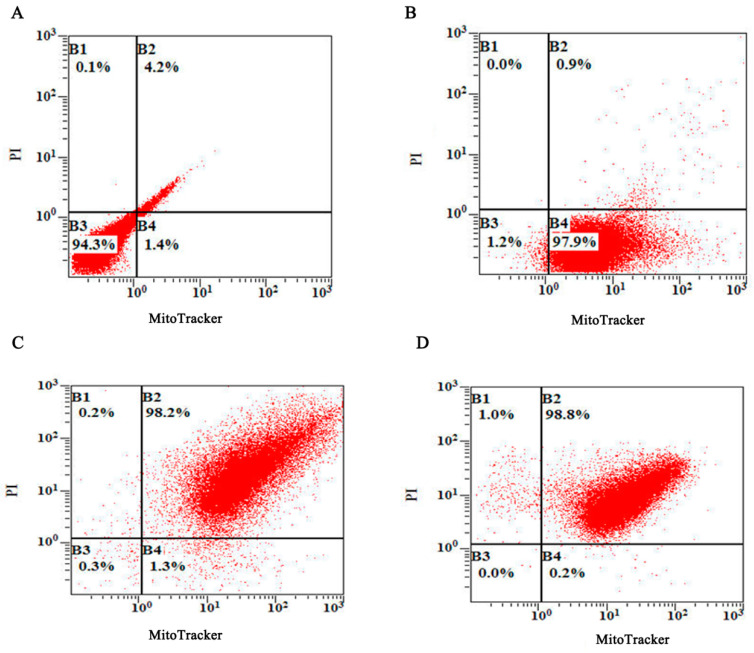
FACS analyses of the green fluorescent MitoTracker-labeled BGs and the red fluorescent PI-labeled plasmid. (**A**) Non-labeled control BGs, (**B**) BGs labeled with MitoTracker, (**C**) MitoTracker-labeled BGs loaded with PI-labeled pGL3 plasmids. (**D**) MitoTracker-labeled BGs loaded with the PI-labeled pVAX1-Ii-*NS3* Th1 plasmid. The *X*-axis shows the green fluorescence intensity of the MitoTracker-labeled BGs plasmid, while the *Y*-axis shows the red fluorescence intensity of the PI-labeled.

**Figure 5 biomedicines-12-00525-f005:**
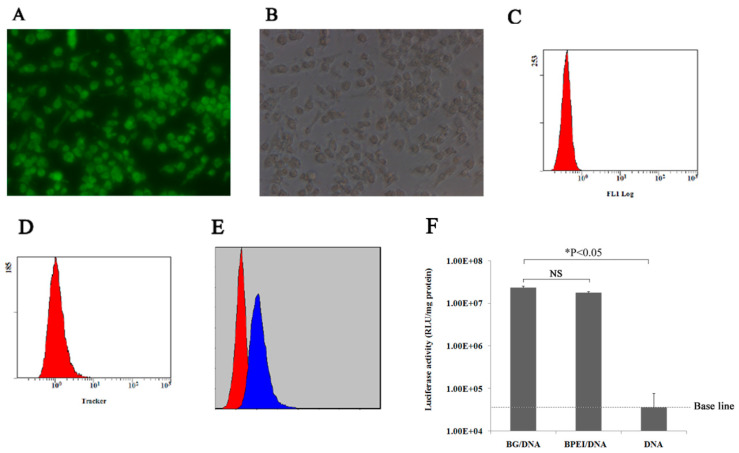
The uptake of bacterial ghosts by RAW 264.7 cells. (**A**) Fluorescent signal in the transfected cells, (**B**) Bright field image of (**A**), The magnification was 24,000×. (**C**,**D**) Fluorescent signal in the control cells and transfected cells, respectively. (**E**) Overlay of the plots for (**C**,**D**). (**F**) Luciferase activity in RAW 264.7 cells at 24 h after transfection. DNA: cells transfected with free plasmid DNA, BG/DNA: cells transfected with BGs loaded with plasmid DNA. PEI/DNA: cells transfected with PEI. Each set of data represents the mean of three independent experiments with error bars indicating the SD.

**Figure 6 biomedicines-12-00525-f006:**
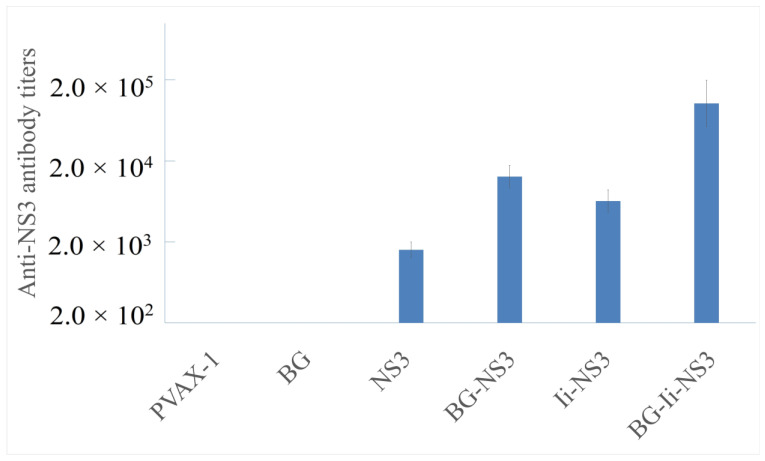
The endpoint titer of anti-NS3 antibodies in vaccinated mice. BALB/c mice were inoculated with BGs loaded with the Ii-*NS3* plasmid (BG-Ii-*NS3*) or with the naked Ii-*NS3* plasmid (Ii-*NS3*). Empty BGs and the pVAX1 plasmid were used for the negative control groups. Each titer value represents the mean of three independent experiments with error bars indicating the SD.

**Figure 7 biomedicines-12-00525-f007:**
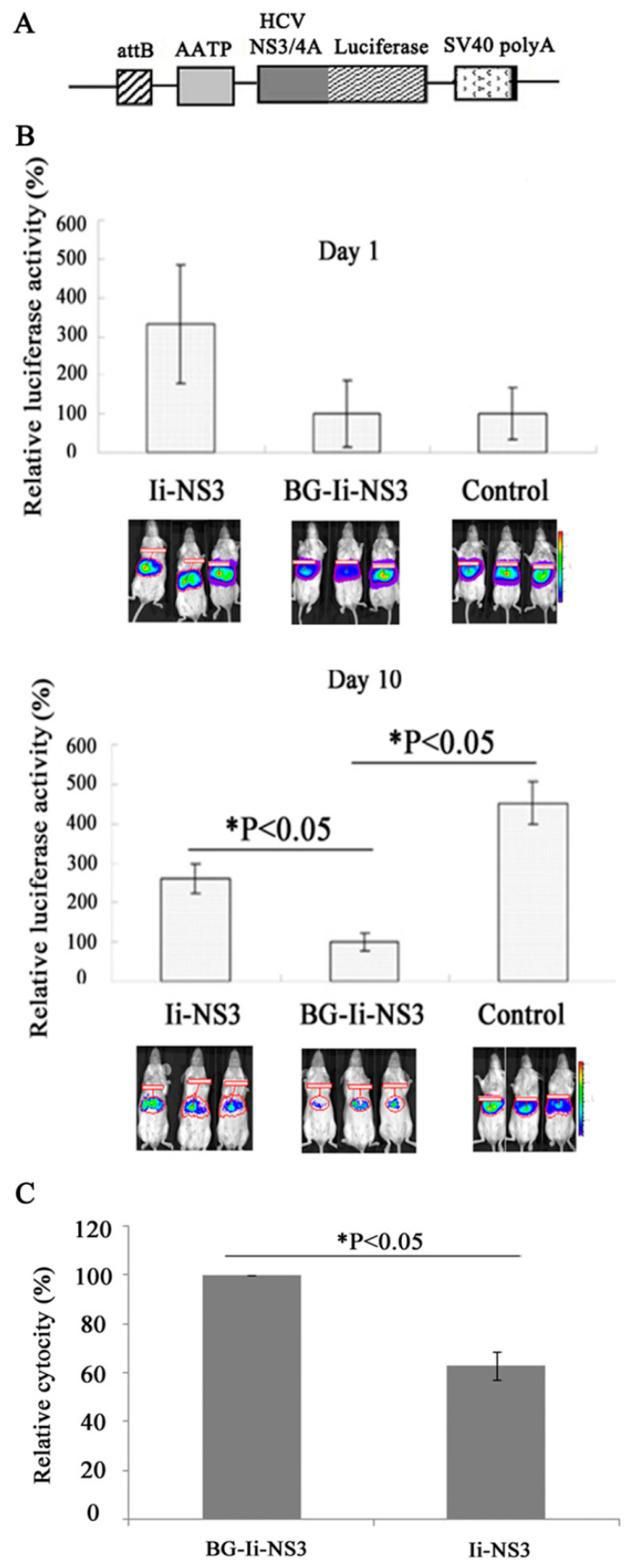
In vivo validation of the immune response. (**A**) Schematic of the construction of the plasmidpAA-*att*B-*NS3*/4A-Fluc. (**B**) Bioluminescence imaging of luciferase expression in BALB/C mice immunized with BG-Ii-*NS3*, Ii-*NS3* or the empty vector, respectively, 1 d and 10 d post-hydrodynamic injection with pAA-*att*B-*NS3*/4A-Fluc and Pphic31φ. (**C**) In vivo induction of T-cell responses against the NS3-restricted epitopes after immunization with Ii-*NS3* or BG-Ii-*NS3*. Splenocytes from BALB/C mice immunized i.p with Ii-*NS3* or with BG-Ii-*NS3* were stimulated with the indicated CTL peptides. The CTL activity of the splenocytes was determined using the cytotox96 non-radioactive cytotoxicity assay. The analyses were conducted 14 days after immunization. Each cytotoxicity value represents the mean of three independent experiments with error bars indicating the SD.

## Data Availability

The data sets used and/or analyzed during the current study are available from the corresponding author upon reasonable request.

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
