# Peer review of "Delivery of a Hepatitis C Virus Vaccine Encoding NS3 Linked to the MHC Class II Chaperone Protein Invariant Chain Using Bacterial Ghosts"

_biomedicines, 2024, doi:10.3390/biomedicines12030525_

Round 1

Reviewer 1 Report

Comments and Suggestions for Authors

This manuscript presents a comprehensive study on the utilization of E. coli bacterial ghosts (BGs) as a novel delivery mechanism for an Ii-linked Hepatitis C Virus (HCV) NS3 DNA vaccine, with the aim of enhancing immune responses. The use of BGs to target antigen-presenting cells (APCs) is a promising approach for the development of DNA vaccines. The work is well-structured, beginning with the preparation and characterization of BGs, followed by in vitro and in vivo evaluations of vaccine efficacy.

These findings suggest that BGs significantly improve transgene expression and antigen presentation in APCs, leading to enhanced humoral and cellular immunity against HCV infection. The comparison of immune responses in BG-Ii-NS3-challenged mice to those treated with Ii-NS3 alone provides convincing evidence for the potential of BG as an effective delivery system. The use of luciferase as a reporter for long-term HCV NS3 expression in vivo is particularly noteworthy as it offers a quantitative measure of the impact of the vaccine on cellular immunity.

Although this study is of considerable interest and potentially impactful for DNA vaccine development, a few areas could benefit from additional clarification or expansion.

1.       The methodology for determining the encapsulation efficiency of plasmid DNA within bacterial ghosts requires further refinement. After loading the bacterial ghosts with plasmid DNA, they were washed to remove any unencapsulated DNA. Dissolving the bacterial ghosts and quantifying the DNA will provide a more accurate measure of encapsulation efficiency. This step is crucial for validating the effectiveness of a bacterial host as a delivery system.

2.       The flow cytometry data presented in Figure 4, specifically the compensation between MitoTracker Green and PI, appear to be incomplete. Incomplete compensation can lead to misleading interpretations of the data. Strict fluorescence compensation protocols should be applied to ensure the accuracy of the flow cytometry results. This is particularly important when multiple fluorophores are used to prevent spectral overlap.

3.       The manuscript compares the induction of antigen-specific antibodies based on absorbance, which may not provide a comprehensive view of antibody functionality or quantity. It would be more appropriate to compare the results using endpoint titers or concentrations to offer a more detailed understanding of the immune response.

4.       There appears to be inconsistency in the methodology used to induce antigen-specific antibodies. While intramuscular administration was used in other parts of the study, Figure 7 unexpectedly employs the hydrodynamic method. This sudden change in methodology is confusing and undermines the comparability of results. An explanation of this methodological shift is necessary to understand the rationale and implications for the study's conclusions.

5.       The manuscript should include information on endotoxin levels within the BG preparations. Given the potential of endotoxins to influence immune responses, quantifying and controlling endotoxin levels is crucial for ensuring the safety and reproducibility of BG-based vaccine delivery systems.

6.       A more detailed discussion of the mechanisms by which BGs enhance immune responses, as demonstrated in this study, would be highly beneficial. This should include potential pathways activated by BGs in APCs, the role of BGs in the enhanced presentation of antigens, and any immunomodulatory effects of the BGs. Understanding these mechanisms is essential for optimizing BG-based delivery systems, and could inform the design of future vaccine formulations.

Addressing these points will further strengthen the manuscript's contributions to the field of immunology and vaccinology, highlighting the potential of BGs as versatile and effective platforms for DNA vaccine delivery.

Author Response

Responses to Decision Letter of Biomedicines

We greatly appreciate your assistance with our manuscript. My coauthors and I are grateful to the referees for pointing out the shortcomings of the manuscript. The referees’ comments are helpful and we have revised the manuscript point by point. All of the corrections were highlighted by red text in the revised manuscript and our responses to Biomedicines Decision Letter (biomedicines-2848911) were summarized point by point as follows:

Comments from the reviewers:

-Reviewer 1

This manuscript presents a comprehensive study on the utilization of E. coli bacterial ghosts (BGs) as a novel delivery mechanism for an Ii-linked Hepatitis C Virus (HCV) NS3 DNA vaccine, with the aim of enhancing immune responses. The use of BGs to target antigen-presenting cells (APCs) is a promising approach for the development of DNA vaccines. The work is well-structured, beginning with the preparation and characterization of BGs, followed by in vitro and in vivo evaluations of vaccine efficacy.

 These findings suggest that BGs significantly improve transgene expression and antigen presentation in APCs, leading to enhanced humoral and cellular immunity against HCV infection. The comparison of immune responses in BG-Ii-NS3-challenged mice to those treated with Ii-NS3 alone provides convincing evidence for the potential of BG as an effective delivery system. The use of luciferase as a reporter for long-term HCV NS3 expression in vivo is particularly noteworthy as it offers a quantitative measure of the impact of the vaccine on cellular immunity.

 Although this study is of considerable interest and potentially impactful for DNA vaccine development, a few areas could benefit from additional clarification or expansion.

 The methodology for determining the encapsulation efficiency of plasmid DNA within bacterial ghosts requires further refinement. After loading the bacterial ghosts with plasmid DNA, they were washed to remove any unencapsulated DNA. Dissolving the bacterial ghosts and quantifying the DNA will provide a more accurate measure of encapsulation efficiency. This step is crucial for validating the effectiveness of a bacterial host as a delivery system.

Response:

Thank you for your comments.

Bacterial ghosts have high DNA loading capacity. Paukner S et al reported that the Up to approximately 6000 plasmids could be loaded per single ghost and the amount of loaded DNA correlated with the DNA concentration used for loading. E. coli ghosts loaded with plasmids encoding the enhanced green fluorescent protein (EGFP) targeted efficiently murine macrophages (RAW264.7) and mediated effective gene transfer.

Paukner S, Kudela P, Kohl G, Schlapp T, Friedrichs S, Lubitz W. DNA-loaded bacterial ghosts efficiently mediate reporter gene transfer and expression in macrophages. Mol Ther. 2005 Feb;11(2):215-23.

  The flow cytometry data presented in Figure 4, specifically the compensation between MitoTracker Green and PI, appear to be incomplete. Incomplete compensation can lead to misleading interpretations of the data. Strict fluorescence compensation protocols should be applied to ensure the accuracy of the flow cytometry results. This is particularly important when multiple fluorophores are used to prevent spectral overlap.

Response:

Thank you for your comments.

To detect BG loaded plasmid DNA by using different fluorescence is inevitably subject to errors, which may be limited by the labeling efficiency of BG and plasmid DNA, or by the systematic errors of the flow cytometer itself. Nevertheless, flow cytometry is recognized as a sensitive and reliable detection method. Our results show that nearly 100% of BG are loaded plasmids, indicating that the method of using BG to load plasmid DNA is mature and reliable. As for the fact that a very small number of BGs without plasmid DNA loading may affect the experimental results, we believe that their impact on the results may be limited.

  The manuscript compares the induction of antigen-specific antibodies based on absorbance, which may not provide a comprehensive view of antibody functionality or quantity. It would be more appropriate to compare the results using endpoint titers or concentrations to offer a more detailed understanding of the immune response.

Response:

Thank you for your comments.

The OD450 value in Figure 6 was detected when the serum sample was diluted 200 times. However,when the sample was diluted to 3200 times, the average OD values of the Ii-NS3 and BG-Ii-NS3 groups were 0.4632 and 1.259, respectively.

If the positive sample OD450/negative sample OD450 is greater than twice as the critical value, the anti-NS3 antibody titer of the BG-Ii-NS3 group (1:102400) was 8 times higher than that of the naked Ii-NS3group (1:12800), indicating that BG cuold efficiently deliver the Ii-NS3 vaccine, greatly improving its immunity. Please see line 311-313.

There appears to be inconsistency in the methodology used to induce antigen-specific antibodies. While intramuscular administration was used in other parts of the study, Figure 7 unexpectedly employs the hydrodynamic method. This sudden change in methodology is confusing and undermines the comparability of results. An explanation of this methodological shift is necessary to understand the rationale and implications for the study's conclusions.

Response:

Thank you for your comments.

The purpose of this project is to establish a dual-target DNA vaccine delivery system. Firstly, Firstly, the DNA vaccine was efficiently delivered to antigen presenting cells by BG to obtain efficient expression.Secondly, the gene fragment encoding NS3 antigen was constructed in the Ii vector, and the expression product was delivered to MHC-II molecule with endogenous targeting, which further improved the antigen presenting ability and immune response level. In order to verify this new strategy of vaccine design, we first evaluate the enhancement effect of dual-target vaccine on immune response level by using mouse intramuscular injection model. The preliminary results confirmed that BG can increase the immune response level of Ii-NS3. However, this model can not evaluate the expression level of DNA vaccine in vivo. Hydrodynamic method and in vivo imaging technology can detect the distribution and expression level of plasmid DNA in mice, and can directly evaluate the delivery effect of plasmid DNA. Therefore, we decided to use hydrodynamic method to evaluate BG delivery efficiency.

  The manuscript should include information on endotoxin levels within the BG preparations. Given the potential of endotoxins to influence immune responses, quantifying and controlling endotoxin levels is crucial for ensuring the safety and reproducibility of BG-based vaccine delivery systems.

Response:

Thank you for your comments.

Due to the presence of many lipopolysaccharides (such as endotoxins) on the surface of Gram negative bacteria, some people are concerned that injection of BG may cause serious toxic side effects. However, Mader et al.have shown that experimental animals have a tolerance level of lipopolysaccharides on the surface of BG at least two orders of magnitude higher than free lipopolysaccharides, and the dosage of BG used in the study does not cause endotoxin-related toxic side effects, thus not affecting its safety as a DNA delivery carrier. Please see ine 396-400.

 A more detailed discussion of the mechanisms by which BGs enhance immune responses, as demonstrated in this study, would be highly beneficial. This should include potential pathways activated by BGs in APCs, the role of BGs in the enhanced presentation of antigens, and any immunomodulatory effects of the BGs. Understanding these mechanisms is essential for optimizing BG-based delivery systems, and could inform the design of future vaccine formulations.

Response:

Thank you for your comments.

 The mechanisms by which BGs enhance immune responses has been added to the article,please see line 366-388.

Addressing these points will further strengthen the manuscript's contributions to the field of immunology and vaccinology, highlighting the potential of BGs as versatile and effective platforms for DNA vaccine delivery.

Response:

Thank you for your comments.

We sincerely hope that our responses to your questions and suggestions about the manuscript will satisfy you. Thanks for your reconsideration of and attention to our manuscript.

Reviewer 2 Report

Comments and Suggestions for Authors

The authors developed DNA vaccine against HCV NS3 using E. coli bacterial ghost and evaluated both in vitro and in vivo systems. Development of HCV vaccine is important and this study will contribute to this. Comments for the authors below:

Major points:

1.     Line 104: I don’t think you can anneal oligos at 95ËšC.

2.     Line 112: I don’t see “NS3 Th1 epitope (lower case)” in the sequence.

3.     Line 174: Please indicate the timing of blood collection.

4.     Line 244: Please explain “PI-labeled plasmid DNA” in section 2.5.

5.     Lines 265-267: X- and Y-axis looks like other way around.

6.     Line 304: Please explain the error bars. SD, SEM?

7.      

Minor points:

1.     Materials and Methods: Please add Cat. No. where applicable so that readers can follow your method easily. Also please explain the statistical method used in this study.

2.     Line168: Please delete “that were 6-8 weeks old”. This is a repetition of Line 164.

Author Response

Responses to Decision Letter of Biomedicines

We greatly appreciate your assistance with our manuscript. My coauthors and I are grateful to the referees for pointing out the shortcomings of the manuscript. The referees’ comments are helpful and we have revised the manuscript point by point. All of the corrections were highlighted by red text in the revised manuscript and our responses to Biomedicines Decision Letter (biomedicines-2848911) were summarized point by point as follows:

Comments from the reviewers:

-Reviewer 2

The authors developed DNA vaccine against HCV NS3 using E. coli bacterial ghost and evaluated both in vitro and in vivo systems. Development of HCV vaccine is important and this study will contribute to this. Comments for the authors below:

Major points:

  1. Line 104: I don’t think you can anneal oligos at 95ËšC.

Response:

Thank you for your comments.

The sentence was modified to “The sense and antisense strands were annealed at 95℃ for 5 min followed by slow cooling to room temperature”, please see line 107 .

  1.  Line 112: I don’t see “NS3 Th1 epitope (lower case)” in the sequence.

Response:

Thank you for your comments.

The DNA sequence NS3 Th1 epitope was revised to lower case, please see line 117 .

  1. Line 174: Please indicate the timing of blood collection.

Response:

Thank you for your comments.

The the timing of blood collection was already showed in the text(line 178-180): “All of the groups received a booster injection at one-week intervals. After the third injection, blood was collected from all of the mice and the serum was obtained.”

  1.  Line 244: Please explain “PI-labeled plasmid DNA” in section 2.5.

Response:

Thank you for your comments.

The plasmid DNA was mixed with PI, stain at 4 ℃ for 30 minutes, then anhydrous ethanol was added a final concentration of 70% to precipitate the plasmid, and  the plasmid was dissolved gain with HBS (above 10mg/ml) for later use.

  1. Lines 265-267: X- and Y-axis looks like other way around.

Response:

Thank you for your comments.

The definition of X-axis and Y-axis has been modified,please see line 271-272 in the text.

  1. Line 304: Please explain the error bars. SD, SEM?

Response:

Thank you for your comments.

Each data represents the mean of three independent experiments with error bars indicating the SD.

Minor points:

  1. Materials and Methods: Please add Cat. No. where applicable so that readers can follow your method easily. Also please explain the statistical method used in this study.

Response:

Thank you for your comments.

The Cat. No. of the main reagents have been added in the materials section.

The statistical methods have been supplemented in the article,

  1. Line168: Please delete “that were 6-8 weeks old”. This is a repetition of Line 164.

Response:

Thank you for your comments.

The words were deleted following the reviewer's advice.

We sincerely hope that our responses to your questions and suggestions about the manuscript will satisfy you. Thanks for your reconsideration of and attention to our manuscript.

Round 2

Reviewer 1 Report

Comments and Suggestions for Authors

Overall, the authors have effectively addressed the concerns by providing comprehensive responses and clarifications for most points. However, it is recommended that the authors present the data on antigen-specific antibodies using endpoint titers instead of absorbance values. This will provide a more precise and detailed understanding of the immune response. Endpoint titers will offer a clearer indication of the quantity of antibodies, which are crucial factors in assessing the efficacy of the vaccine.

Author Response

Responses to Decision Letter of Biomedicines

We greatly appreciate your assistance with our manuscript. My coauthors and I are grateful to the referees for pointing out the shortcomings of the manuscript. The referees’ comments are helpful and we have revised the manuscript point by point. All of the corrections were highlighted in the revised manuscript and our responses to were summarized as follows:

Comments and Suggestions for Authors

Overall, the authors have effectively addressed the concerns by providing comprehensive responses and clarifications for most points. However, it is recommended that the authors present the data on antigen-specific antibodies using endpoint titers instead of absorbance values. This will provide a more precise and detailed understanding of the immune response. Endpoint titers will offer a clearer indication of the quantity of antibodies, which are crucial factors in assessing the efficacy of the vaccine.

Response:

Thank you for your comments.

Method of endpoint titer calculation was added; please see Line 228-229.

The humoral immune responses results have been changed to endpoint of immunized mice, please see line 306-309 and in the new revised Figure 6. The title of Figure 6 has also been changed accordingly.

We sincerely hope that our responses to your questions and suggestions about the manuscript will satisfy you. Thanks for your reconsideration of and attention to our manuscript.